# Barocrinology: The Endocrinology of Obesity from Bench to Bedside

**DOI:** 10.3390/medsci8040051

**Published:** 2020-12-21

**Authors:** Sanjay Kalra, Nitin Kapoor, Saptarshi Bhattacharya, Hassan Aydin, Ankia Coetzee

**Affiliations:** 1Bharti Hospital, Karnal 132001, India; 2Christian Medical College Vellore, Vellore 632004, India; nitin.endocrine@gmail.com; 3NCD Unit, Melbourne School of Population and Global Health, University of Melbourne, Melbourne 3004, Australia; 4Department of Endocrinology, Max Superspeciality Hospital, Patparganj 110092, India; saptarshi515@gmail.com; 5Department of Internal Medicine, Yeditepe University Hospital, 34755 Istanbul, Turkey; hsnydn@yahoo.com; 6Division of Endocrinology, Stellenbosch University, Cape Town 8000, South Africa; blommeland@gmail.com

**Keywords:** bariatric surgery, weight loss, endocrine changes after bariatric surgery, endocrine changes after weight loss

## Abstract

Obesity has reached pandemic proportions. Hormonal and metabolic imbalances are the key factors that lead to obesity. South Asian populations have a unique phenotype, peculiar dietary practices, and a high prevalence of consanguinity. Moreover, many lower middle-income countries lack appropriate resources, super-specialists, and affordability to manage this complex disorder. Of late, there has been a substantial increase in both obesity and diabesity in India. Thus, many more patients are being managed by different types of bariatric procedures today than ever before. These patients have many types of endocrine and metabolic disturbances before and after bariatric surgery. Therefore, these patients should be managed by experts who have knowledge of both bariatric surgery and endocrinology. The authors propose “Barocrinology”, a novel terminology in medical literature, to comprehensively describe the field of obesity medicine highlighting the role of knowing endocrine physiology for understating its evolution, insights into its complications and appreciating the changes in the hormonal milieu following weight loss therapies including bariatric surgery. Barocrinology, coined as a portmanteau of “baro” (weight) and endocrinology, focuses upon the endocrine and metabolic domains of weight physiology and pathology. This review summarizes the key pointers of bariatric management from an endocrine perspective.

## 1. Introduction

Obesity has become a global pandemic, with numbers nearly tripling since 1975 [1]. Initially thought to be a disease of the developed world, it has now shown to be a rapidly increasing trend even in the Indian subcontinent [2]. The World Health Organization (WHO) defines obesity and overweight as an “abnormal or excessive fat accumulation that presents a risk to health” [3].

Obesity is characterized by several hormonal disturbances. These can be primary abnormalities which can contribute to the development of obesity. Alternatively, there can be secondary abnormalities which can often be reversed by weight loss. Obesity is also the symptom of many endocrine disorders, and treating these disorders can cause weight loss.

Obesity has conventionally been managed as a disease that results from an imbalance between energy intake and energy expenditure. However, it is now understood as a “neurohormobehavioral” disease, resulting from altered hypothalamic control of hunger, satiety, and energy expenditure [4,5]. Additionally, it has monogenic (single mutated gene), syndromic (a group of genes are involved), and polygenic (common obesity with multiple genes involved) components affecting the endocrine system [6]. These are even more prominent in consanguineous populations, like South Asians [7]. Knowledge of endocrine physiology and dysfunction is therefore necessary to manage obesity.

Obesity is managed by pharmacological, non-pharmacological, and surgical means. Bariatric surgery is being increasingly used in lower middle-income countries (LMICs) like India to treat obesity. Patients who undergo bariatric surgery need pharmacological and non-pharmacological (diet, medical nutrition therapy [MNT], and exercise) support before and after surgery. Therefore, these patients should be managed by experts who have knowledge of both bariatric surgery and endocrinology. These endocrinologists may be referred as “Barocriniologists” (having expertise in endocrine management of bariatric surgery patients).

Barocrinology is defined as the science of weight metabolism and obesity-related disorders. Coined as a portmanteau of “baro” (weight) and endocrinology, barocrinology focuses upon the endocrine and metabolic domains of weight physiology and pathology. It includes, in its ambit, the endocrine and metabolic, etiopathogenesis, clinical presentation, therapy, expected benefits, and possible side effects of therapy, as well as the monitoring and follow-up of obesity.

However, in LMICs like India, trained Barocriniologists are not widely available in many health settings. Hence, patients with obesity are usually evaluated and treated by their primary care physicians (PCPs). Even the patients, who undergo bariatric surgery, go back to their PCPs or endocrinologists post-surgery for follow-ups.

Hence, this review is likely to help PCPs and endocrinologists better manage obesity in patients undergoing bariatric surgery, as it incorporates the close association between obesity and the endocrine system. This review covers the endocrine causes of obesity and the effect of pharmacological treatment, bariatric surgery, and non-pharmacological treatments on the endocrine system. It also covers practical steps for evaluating and managing obesity from an endocrine perspective.

## 2. Endocrine Causes of Obesity

Obesity can be the result of structural lesions at the hypothalamic–pituitary region or a consequence of inherited and genetic syndromes. These abnormalities cause hypothalamic obesity. Alternatively, functional neuroendocrine hormone abnormalities can cause obesity [8].

### Syndromic Obesity

Chromosomal abnormalities and genetic mutations can cause obesity. This inheritance can be monogenic (single mutated gene) or syndromic (a group of genes are involved) [6]. Syndromic obesity is a term used to describe obesity that is inherited in an autosomal or an X-linked pattern (Table 1). Patients with syndromic obesity can present with dysmorphic features, mental retardation, hyperphagia, abnormal body organs, and/or other features of hypothalamic dysfunction [6]. Obesity in such individuals is usually early-onset, often with a history of consanguinity in the family [6,7].

## 3. Endocrine Hormone Levels in Obesity and Weight Loss

Endocrine hormone changes in obesity and after achieving weight loss are shown in Table 2 [12,13,14].

## 4. Endocrine Evaluation of obesity

### 4.1. Endocrine Condition: Laboratory Evaluation: Do’s and Don’ts

The European Society of Endocrinology (ESE) Clinical Guideline on the Endocrine Work-up in Obesity (2020) gives a detailed framework for laboratory evaluation of endocrinopathies in a patient with obesity, as outlined in Table 3 [13,16,17,18].

### 4.2. Anthropometric Evaluation in Endocrinopathy

Anthropometric evaluation is a part of detailed clinical and systemic examination of a patient presenting with obesity. Body mass index (BMI) is the most commonly utilized tool for measuring obesity. It is a measure of a person’s weight in kilograms divided by the square of the height in meters. The BMI classification obesity for adults by the World Health Organization and in Asian sub-continents is given in Table 4.

Though BMI is more accurate than skin-fold measurements in community settings, and provides a reasonable estimate of body fat, it has certain limitations. BMI does not distinguish between lean mass and fat mass, nor throws light on fat distribution. The literature shows that body fat distribution and not excess weight is associated with many obesity-related risk factors [21,22,23,24]. Accurate body fat estimation techniques like computed tomography or magnetic resonance imaging are very costly and require sophisticated equipment [25]. Hence, these cannot be routinely used in community settings. Therefore, along with BMI assessments for classifying obesity, waist circumference (WC) and waist–hip ratios (WHR) can be measured to assess body fat distribution (subcutaneous vs. visceral) [13,24,26,27]. Visceral fat, and not the subcutaneous fat (Figure 1) is comparatively more significantly linked with metabolic syndrome [24]. Table 5 shows the predictive value of BMI, WC, and WHR in capturing subcutaneous fat [13,28].

Moreover, a recent study in the Indian population revealed that WC, WHR, and waist:height ratio (WHtR) were the most appropriate obesity indicators that predict T2DM. WC, WHR, and WHtR values were significantly higher in patients with T2DM than in those without (*p* < 0.001). The receiver-operating characteristic (ROC) curve analyses showed that these three obesity indicators identified well with T2DM. The ROC for WHR was 0.67 (95% confidence interval [CI] 0.59, 0.75), WHtR was 0.66 (95% CI 0.57, 0.75), and for WC, was 0.64 (95% CI 0.55, 0.73) [30].

Furthermore, certain surrogate measures of visceral adipose tissue estimation, like METS-VF, have also been validated, and can be useful adjuncts to predict VAT content in resource-limited settings [31].

### 4.3. Approach to Evaluating Syndromic Obesity

Since patients with syndromic obesity have characteristic dysmorphic features, a detailed family history and clinical examination is the starting point of evaluation of syndromic obesity. Endocrine abnormality and genetic defects are tested in cases highly suspicious of a particular syndrome (Figure 2) [32].

## 5. Endocrine Impact of Obesity Management

### 5.1. Endocrine Impact of Non-Pharmacological Management of Obesity

Non-pharmacological management of obesity includes lifestyle changes, such as diet and exercise. Stress in everyday life is known to be associated with endocrine responses that contribute to obesity. Hence, stress management is an important factor in the endocrine management of obesity.

#### 5.1.1. Potential Endocrine/Metabolic Impact of Exercise

Exercise impacts all the endocrine glands. During exercise, the pituitary releases growth hormones, the thyroid releases T4, and the adrenals release cortisol and aldosterone. All these hormones increase the body’s capacity to exercise, stay alert, focus on the high-intensity task, provide energy for the exercise, and prevent dehydration (especially aldosterone). Exercise releases endorphins which help reduce anxiety and stress. Exercise also increases release of testosterone [33].

The level and type of hormone differs with the type and stage of exercise. Together, the hormones increase lipolysis, peripheral utilization of glucose, increase protein synthesis and muscle-building (in strength training exercises), and bring about the beneficial effects of increased body energy and weight loss, especially in individuals with obesity [34,35].

Exercise also increases insulin sensitivity, which helps in utilization of glucose and reduces postprandial insulinemia. Timing of nutrients is important for bringing out the best results of an exercise. A study found that whole-body and skeletal-muscle lipid utilization increased if an exercise was done before a mixed-macronutrient meal, including 65% of kcal carbohydrate versus exercise, was performed after taking the meal. The lipid utilization increased two-fold when this exercise was continued for six weeks [36].

Hence, all patients undergoing bariatric surgery should be encouraged to take up physical activity. A proper exercise plan should be built into their routine as per their capability. Exercise should include both aerobic exercises and strength training. Exercise also helps release endorphins, which help combat stress. The various endocrine and metabolic benefits of exercise are shown in Figure 3.

More recently, emerging literature has also highlighted the role of skeletal muscle as a secretory organ and its role in the regulation of body fat metabolism. Myokines induced by physical activity and muscle contraction have paracrine effects on the muscle. They promote myogeneisis and muscle hypertrophy in the individual. Beyond the muscle, they also assist in promoting glucose uptake, lipolysis, β-oxidation of fat, angiogenesis, and revascularization. Muscular irisin, β-aminoisobutyric acid (BAIBA), and fibroblast growth factor 21 (FGF21) may also lead to conversion of white adipose tissue to brown adipose tissue. Overall, they improve lipid and glucose profiles in the individual [37], a potential endocrine/metabolic of impact of stress management

A patient going for bariatric surgery is under a lot of stress. Bariatric surgery, like any other surgery, induces stress in the body. Any stressful situation triggers the hypothalamic-pituitary-adrenal (HPA) axis to release hormones to combat this stress. The corticotrophin-releasing hormone (CRH) is then released from the hypothalamus, which activates the anterior pituitary to release ACTH, which in turn stimulates the adrenal cortex to release glucocorticoids. Stimulation of the HPA also releases catecholamines and vasopressin [38]. There is downregulation and thus a decrease in circulating gonadotropins and gonadal steroid hormones, T3 and T4. Growth hormone release increases, the prolactin level can either increase or decrease, and the insulin level may decrease [38].

As a net result, there is poor glycemic control and preponderance of central fat deposition. The secretion of orexogenic hormone ghrelin increases, resulting in increased appetite and food intake [39]. Therefore, proper stress management is important to avoid these obesity-contributing factors before and after bariatric surgery. Proper counseling and an addressing of ways to combat stress should be sought from a trained psychologist.

#### 5.1.2. Potential Endocrine/Metabolic Impact of Diet

Calorie restriction is the main component of weight-loss dieting. This calorie restriction impacts the endocrine system. There is a fall in T_3_ and gonadal function, and an increase in cortisol secretion [40]. Low-calorie diets aid T2DM reversal and improve the lipid profile [41].

The Palaeolithic diet (low carbohydrate; lean protein constituting 30–35% daily caloric intake; 45–100 g fibre diet from plant-based non-cereal sources) has beneficial effects on weight, metabolic syndrome, insulin secretion, glycaemia and glucose tolerance, lipid profiles, adipo-cytokine profiles, and cardiovascular risk factors [42].

A ketogenic diet (high-fat, low-carbohydrate, with adequate proteins) results in higher glucagon and cortisol levels before, during, and after aerobic exercise. Glucagon levels and, to a lesser extent, cortisol levels were associated with a more than two-fold increased peak fat oxidation rate [43].

#### 5.1.3. Medical Nutrition Therapy (MNT)

Diet therapy in obesity may include MNT/formula MNT. The metabolic goals of MNT can vary, and include the lowering of blood pressure, serum cholesterol, and/or fasting blood glucose [44]. In patients with obesity who have an endocrine disease, planned diet restrictions are aimed at improving the gland function.

The landmark Diabetes Remission Clinical Trial (DiRECT) included patients with T2DM and BMIs of 27–45 kg/m^2^ in intervention and control groups (*n* = 149 each; intent to treat). In the intervention group, the total diet was replaced with a formula diet of 825-853 kcal/day for 3–5 months. Their antidiabetic and antihypertensive drugs were withdrawn, and food was carefully stepped up and reintroduced over a period of 2–8 weeks. The control group did not receive formula MNT. All patients received structured support for maintaining long-term weight loss. At 12 months, weight loss of 15 kg or more was recorded in 24% and 0% of participants in the intervention and control groups, respectively (Fisher’s exact *p* < 0.0001). Diabetes remission was achieved in 46% and 4% of participants in the intervention and control groups, respectively (odds ratio 19.7, 95% CI 7.8–49.8; *p* < 0.0001) [45]. Since the DiRECT trial included patients with T2DM with characteristics similar to patients in routine practice, the results of the trial can be generalized to a wider population [46].

Another trial showed that structured MNT improves glycemic and cardiovascular parameters. Three groups of patients received MNT from a registered nutritionist (RDN). Group A received an individualized eating plan, and Groups B and C received a structured meal plan; Group C also received weekly phone support from RDN. After 16 weeks, there was no change in HbA1c from baseline in Group A, but HbA1c decreased significantly in Groups B and C (−0.66%, 95% CI -1.03 to −0.30 and −0.61%, 95% CI −1.0 to −0.23, respectively; *p* < 0.001). Groups B and C also showed significant reductions in body weight and cardiovascular parameters, such as body fat percentage and waist circumference [47].

The effect of long-term calorie-restricted diets (CR) with adequate protein and micronutrient intake on thyroid function was compared with age- and sex-matched sedentary (WD), and body-fat-matched exercising (EX) subjects (*n* = 28 each; all subjects were healthy lean and weight-stable). The CR group had lower serum T3 concentration than the WD and EX groups (*p* ≤ 0.001). However, serum TSH, total and free T4 (fT4), and reverse T3 concentrations were similar among groups [48]. Another study showed that a low-calorie diet program was associated with a significant increase in free T3, fT4, and significant decrease in TSH, leptin, and BMI compared to baseline [49].

Moderate dietary restrictions in 47 subjects with BMIs of 25–45 kg/m^2^ resulted in weight loss of 6.3 ± 0.9 kg (6.5 ± 1.0%). TSH and T3 concentrations correlated significantly with fat mass (*p* = 0.024 and *p* = 0.005, respectively) at baseline. T3 decreased significantly after weight loss (*p* < 0.001) but there were no significant changes in TSH or fT4. There was a significant correlation between decrease in serum T3 and decrease in weight (*p* < 0.001). There was a significant decrease in T3:fT4 ratio (*p* = 0.02) in subjects who lost >5% of body weight [50].

### 5.2. Endocrine Impact of Pharmacological Management of Obesity

#### 5.2.1. Endocrine Impact of Weight-Reducing Drugs

There are six major anti-obesity medications approved by the U.S. Food and Drug Administration (FDA): orlistat, phentermine, phentermine-/topiramate-extended release (ER), lorcaserin-, liraglutide-, and naltrexone-sustained release (SR)/bupropion SR (only injectable) (Table 6). Most of these drugs either reduce appetite or enhance satiety, except orlistat, which works by decreasing fat absorption [51]. In patients with obesity with T2DM, glucagon-like peptide-1 receptor agonists (GLP-1 RA), like semaglutide and dulaglutide, help control hyperglycemia and reduce weight. GLP-1 RAs act on alpha cells of the pancreas to inhibit glucagon secretion, and have extra-pancreatic effects, such as increased satiety and delayed gastric emptying [52]. Thus, their impact is mainly on the gut endocrine system, and is similar to but less than that achieved by bariatric surgery. Further details are covered under bariatric surgery.

#### 5.2.2. Weight Impact of Endocrinotropic Drugs

Many medications used to treat comorbidities have an endocrinotropic effect, which means they act on the endocrine system. Hence, these drugs can contribute to hyperglycemia, and thus T2DM [53], dyslipidemia, and body fat redistribution that favors central and visceral obesity with or without subcutaneous fat atrophy (lipodystrophy) [54], non-alcoholic steatohepatitis (NASH), and metabolic syndrome [55].

For treating co-morbid conditions during the pre- and post-bariatric surgery period, clinicians should cautiously choose medications that will be used over a long period and have weight gain as a side-effect. Common medications likely to cause weight gain include antipsychotics, antidiabetics, antihypertensives, antidepressants, and antihistaminics. Medications for co-morbid conditions that are either weight-neutral or have weight loss as a side effect should be preferred over medications that have weight gain as a side effect (Table 7) [56,57]. However, these medications prescribed for co-morbid conditions that have weight loss as a side effect should not be solely prescribed with the intention of losing weight.

### 5.3. Endocrine Impact of Bariatric Surgery

The anatomic changes post-bariatric surgery result in reduced appetite, reduced food consumption, and change in gut hormone secretion. Together, these changes profoundly improve the overall systemic metabolic profile [58]. Apart from dramatic weight loss benefits, bariatric surgery significantly improves insulin resistance and leads to resolution of type 2 diabetes in many cases [59]. Other benefits of bariatric surgery include reduction in other comorbidities, such as osteoarthritis, obstructive sleep apnea, respiratory dysfunction, gonadal dysfunction, and improvement in symptoms of polycystic ovarian syndrome (PCOS) in women and androgen deficiency in men, known as male obesity-associated secondary hypogonadism (MOSH) [60,61,62,63], improvement in physical function and quality of life [64], as well as the lowering of triglyceride levels, improvement in HDL-C levels, and resolution of cardiovascular risk factors [65,66,67,68,69,70,71,72].

#### 5.3.1. Impact of Bariatric Surgery on Islet Function, Insulin Secretion, and Glucose Control

All bariatric surgery procedures modify the gastrointestinal tract in different ways, but all procedures improve the status of T2DM. This reduction in hyperglycemia that is achieved post-surgery is superior to that achieved through medical and lifestyle management [73].

Of the various bariatric procedures (Figure 4), studies show that biliopancreatic diversion (BPD) gives the most benefits in T2DM remission, sustained weight loss, and improved lipid profile. However, this is a complex procedure and has more post-operative complications than other procedures, and hence, is less used. Of the commonly used procedures, Roux-en-Y gastric bypass (RYGB) and sleeve gastrectomy (SG) provide similar benefits for T2DM remission, metabolic profile, and weight loss [74]. The glucose control post-BPD, -RYGB, and -SG is independent of weight loss, but improves significantly with loss of weight. The glucose control after laparoscopic adjustable gastric banding (LAGB) is similar to that achieved through dietary restrictions, and is less than that achieved through BPD, RYGB, and SG [73].

Enteral signals after bariatric surgery stimulate the islet β-cells to secrete insulin. Additionally, postprandial GLP-1 secretion increases after RYGB or SG (Table 6), which in turn further enhances insulin secretion [73]. In addition, insulin sensitivity improves after RYGB, SG, BPD, and LAGB.

#### 5.3.2. Hormonal Impact of Bariatric Surgery on Hypoglycemia

Hypoglycemia is a serious complication after bariatric surgery. After gastric bypass, there is a marked decrease in levels of glucagon, cortisol, and catecholamine in response to hypoglycemia. The sympathetic nerve response to hypoglycemia also decreases. The growth hormone response is delayed, but its peak level is higher than seen pre-surgery. Though GLP-1 and gastric inhibitory polypeptide levels rise during hypoglycemia, the response is lower compared to the pre-surgery level. Therefore, gastric bypass results in re-wiring of glucose homeostasis, which reduces neurohormonal responses to hypoglycemia, and therefore, the symptoms of hypoglycemia [75]. Post-bariatric surgery hypoglycemia is therefore difficult to diagnose, and clinicians should remain alert about its possibilities.

#### 5.3.3. Impact of Bariatric Surgery on Gut Hormones: Hunger and Satiety

Significant changes in gut hormones are brought about by RYGB, SG, BPD, duodenal switch (DS), LAGB, and jejunoileal bypass. These changes in gut hormones positively stimulate satiety and reduce hunger, and are thought to be responsible for the metabolic improvements seen post-bariatric surgery (Table 8) [73,76,77,78].

#### 5.3.4. Impact of Bariatric Surgery on Energy Expenditure

Bariatric surgery induces weight loss. However, this weight loss may disproportionately reduce the resting energy expenditure (REE) and fat-free mass (FFM; trunk organ mass, skeletal mass) which may predispose these patients to weight regain and sarcopenia. A study showed significant REE (mean ± standard error) differences between patients who had RYGB versus controls were 43.2 ± 34 kcal/day (*p* = 0.20) at one year; (−)27.9 ± 37.3 kcal/day (*p* = 0.46) at the second year; and 114.6 ± 42.3 kcal/day (*p* = 0.008) at year five. Compared to controls, patients who had RYGB had greater trunk organ mass (~0.4 kg) and less skeletal mass (~1.34 kg) at each visit [79].

Another study compared the REE and thermic effect of food (TEF) between adolescent females who underwent SG versus controls. The study found that fasting REE and post-meal TEF did not differ between the groups. However, compared to controls, patients who underwent SG consumed less daily energy during their usual food intake, had lower postprandial glucose, but higher insulin and C-peptide [80].

#### 5.3.5. Impact of Bariatric Surgery on Somatotropic Axis: Growth Hormone (GH)/Insulin-Growth Factor-1 (IGF-1)

The literature shows that obesity-related alterations in the somatotropic axis (Table 1) are restored after bariatric surgery. GH secretion has been found to increase significantly after BPD [81]. Partial recovery of the somatotropic axis has been reported after RYGB (a procedure with both malabsorptive and restrictive components) [82,83]. There is about a 2.7-fold increase in GH after malabsorptive procedures (BPD, DS), as compared to a 1.4-fold increase after restrictive procedures (LAGB), with similar benefits in cardiovascular profile and BMI reduction. Hence, malabsorptive procedures may be better suited for patients with somatotropic deficiencies [82,84]. However, further research involving large trials is necessary to confirm this benefit.

Another study showed that the GH and IGF-1 levels that were compromised in patients with severe obesity were increased after RYG at 6 months in women, and at 12 months in both men and women. BMI reduced for both men and women at 6 months, and the reduction improved by 12 months post-surgery [85].

In patients with obesity who underwent LAGB, the GH response to stimulus (GHRH with arginine) before and after surgery was found to be significantly associated with body composition. Weight loss and improvement in body composition was higher in patients with somatotropic axis alterations who recovered their GH response to stimuli after surgery [86,87]. These benefits were also seen in patients who had no somatotropic axis alterations. Hence, alterations in the somatotropic axis may be a useful tool to assess the patient’s response to bariatric surgery [87]. However, GH should not be used to treat obesity in patients with normal GH levels.

IGF-1 secretion usually shows a slower response after bariatric surgery, even though conflicting reports have been published [84,88]. The response is similar to that seen in weight loss achieved through non-surgical means. This could be due to the catabolic status induced by bariatric surgery because of the malabsorptive and diet-/calorie-restrictive effects [85,89].

#### 5.3.6. Impact of Bariatric Surgery on Hypothalamic/Pituitary/Gonadal Axes in Women

Women with obesity can have sub-fertility, or infertility. Though PCOS is the most common gonadal dysfunction in women with obesity, idiopathic hypogonadotropic hypogonadism is also associated with increased prevalence of obesity [63,90].

A study showed that sexual dysfunction, as assessed by the Female Sexual Function Index score, improved in about 68% of women with obesity, but this improvement was not dependent on surgery type or amount of weight loss achieved [91]. Bariatric surgery helps restore the disturbed levels of gonadotropin-releasing hormones (GnRH)/follicle-stimulating hormones (FSH), luteinizing hormones (LH)/testosterone, and estradiol (Table 1) to near normal levels.

Abdominal adiposity in women with PCOS results in a vicious circle where visceral fat promotes formation of excess androgen from the ovaries and adrenals, and the excess androgens in turn favor body fat deposition. Women who have excess androgen secretion due to a primary defect in steroidogenesis are predisposed to androgen excess disorders. These women are also predisposed to developing PCOS in response to visceral fat accumulation and obesity [92]. Therefore, women who do not have this primary defect in steroidogenesis do not develop PCOS, even if they have extreme obesity or insulin resistance [93].

Gonadal dysfunction in women with PCOS improves, and may even resolve with weight loss [94]. A meta-analysis showed that the pre- to post-operative PCOS status improved from 45.6% to 6.8% at one year post-bariatric surgery [95]. This improvement in PCOS status is brought about by weight loss achieved through bariatric surgery, and maintained through medication and lifestyle modification. Studies show that this improvement post-bariatric surgery correlates with a decrease in serum total testosterone, androsterone, and sulfate dehydroepiandrosterone (DHEA) with resolution of menstrual irregularities and hirsutism. Serum estradiol, FSH, and SHBG levels also improve (Table 1) [14,96,97,98]. In the study by Sarwer et al. (2014), women who underwent RYGB or LAGB showed significant improvement in sexual desire and function, along with improvement in body image and decrease in depressive symptoms. However, results from these studies should be weighed cautiously as the definition of PCOS varies from study to study, making comparisons difficult.

Women who have PCOS also face issues of sub-fertility or infertility. However, studies looking into the improvement in fertility with bariatric surgery are scarce. In a study, 24 women with PCOS and obesity who achieved weight loss post-RYGB saw an improvement in menstrual irregularities and hirsutism. Additionally, five patients who could not conceive pre-RYGB, conceived after surgery [99].

In 20 women with PCOS and gonadal dysfunction who underwent RYGB, 82% saw an improvement in gonadal dysfunction with decrease in menstrual irregularities in most patients, and resolution of hirsutism (29%). About half the patients who could not conceive before RYGB, could conceive post-surgery [100].

In female patients with PCOS and metabolic features, metformin should be started. Metformin should not be started with the sole aim to reduce body weight. Similarly, estrogen should not be started in postmenopausal obese women with the sole aim to reduce body weight [94].

Hence, women with grade II or III obesity, who face infertility despite going through a structured weight loss program for six months, may benefit from bariatric surgery. However, pregnancy should be avoided preferably for a year post-surgery and at least six months post-surgery, as this is the time for rapid weight loss and significant changes in the endocrine hormones. Patients who have undergone bariatric surgery are also prone to nutritional deficiencies. Hence, pregnancy should be carefully monitored by a specialized multidisciplinary team that should include a Barocraniologist/Endocrinologist [101].

#### 5.3.7. Impact of Bariatric Surgery on Hypothalamic/Pituitary/Gonadal Axes in Men

Excess adipose tissue in obese men is likely to contribute to androgenic deficiency through inhibition of pituitary gonadotropin. MOSH is characterized by low serum testosterone, inappropriately low or normal LH and FSH, and high relative estradiol levels [102]. Abdominal adiposity confers a vicious cycle in the development of MOSH by inhibiting hypophyseal gonadotropin secretion, with the resulting androgen deficiency promoting body fat accumulation with a decrease in muscle and lean mass [103,104].

Weight loss after bariatric surgery, such as RYBG and SG, improves sexual function, as depicted by an improved score on the International Index of Erectile Function. The levels of total testosterone, FSH, and prolactin also improve [105,106].

In patients with obesity who have biochemical and clinical hypogonadism, weight loss should be encouraged to restore eugonadism. Testosterone may be tried if features of clinical and biochemical hypogonadism persist despite weight loss. However, testosterone should be started after other causes of hypogonadism are ruled out. Testosterone should also never be offered as the first therapeutic measure in a male patient seeking fertility treatment. Testosterone should not be started with the sole purpose of losing weight. Efforts should be made to monitor testosterone levels to keep them in the normal age related range. If even after 6-12 months of testosterone therapy, there are biochemical benefits but no clinical benefit, then testosterone should be stopped [94].

#### 5.3.8. Impact of Bariatric Surgery on Bone Metabolism

Bone metabolism in obesity is quite complex. Vitamin D deficiency in obesity has been linked to secondary hyperparathyroidism [107].) Paradoxically, patients with obesity have higher bone mineral content (BMC) and bone mineral density (BMD), and these therefore may have a protective role in osteoporosis [107].

Similarly, contradictory hormonal effects are seen after bariatric surgery. PYY deficiency is associated with low BMD [108]. However, PPY levels increase after RYGB, LAGB, SG, and BPD. GLP-1 levels rise after BS procedures such as BPD, but this procedure is less used. Exogenous administration of GLP-1 and GLP-2 after bariatric surgery may improve BMD [109]. Adipokines, such as leptin, decrease after bariatric surgery, and positively affect bone metabolism [110]. On the contrary, adiponectin, which increases after bariatric surgery, has been negatively correlated with BMD [111].

However, nutritional deficiencies after bariatric surgery play a major role in bone metabolism. Deficiency of vitamin B_12_, folate, iron, calcium, vitamin D, and zinc are commonly seen. Further bariatric surgery further reduces vitamin D levels and BMD, enhancing bone resorption and hyperparathyroidism [112]. Therefore, patients undergoing bariatric surgery have higher post-surgical risk of fracture [113].

Adequate calcium and vitamin D supplementation should be ensured in patients before and after bariatric surgery. Exercise as per capacity and exposure to sunlight also helps improve vitamin D levels.

#### 5.3.9. Impact of Bariatric Surgery on Thyroid Function

Bariatric surgery improves the thyroid function in patients with obesity and hypothyroidism. This improvement of thyroid function is accompanied by a reduction in dosages of thyroid medication. The mean TSH decline is directly related to the baseline TSH value and does not correlate with BMI reduction. This shows that the improvement in thyroid function has a hormonal component and is not related to weight loss [114,115]. A study from China showed that decrease in leptin levels was associated with improved thyroid function in patients with obesity, T2DM, and euthyroidism [116].

In 83 patients with obesity and hypothyroidism treated with laparoscopic SG (LSG; >80%) or RYGB, there was a significant decrease in mean BMI and thyroid-stimulating hormones (TSH) in the 6 and 12 months post-surgery period. The fT4 levels remained stable. A year after surgery, 13.2% of patients did not require thyroid hormone replacement (THR) therapy. THR doses were reduced significantly in the rest of the patients (*p* < 0.02) [114]. Rudnicki et al. (2018) reported similar results in 90 patients treated with either LSG or RYGB. In their observational study of 1581 patients with hypothyroidism and obesity mainly treated with LSG and RYGB, Matute et al. (2018) also observed a significant reduction in BMI (*p* < 0.0001) and THR doses (*p* = 0.071) after one year post-surgery. The mean TSH measurement (*p* = 0.153) and the mean thyroxine measurement (*p* = 0.021) improved after one year post-surgery [117].

Overt hypothyroidism (elevated TSH and decreased FT4) in patients with obesity is treated irrespective of the antibodies’ status. However, thyroid hormones should not be used for treating obesity if the thyroid function is normal. Hyperthyrotropinaemia (elevated TSH and normal FT4) in obesity should not be treated with the sole aim of reducing body weight. The TSH level (Table 9), thyroid antibodies, and age should be considered when deciding whether or not to treat hyperthyrotropinaemia in obesity [13].

#### 5.3.10. Impact of Bariatric Surgery on Adrenal Function

Adrenal insufficiency is a common complication after bariatric surgery. Therefore, clinicians should be very vigilant about symptoms and signs of adrenal insufficiency after bariatric surgery, such as anxiety, sweating, transient loss of consciousness, hypoglycaemia, and conflicting symptoms of profound weight loss or weight regain [119,120]. However, in patients who have rapid weight loss after bariatric surgery, these symptoms are often masked.

The possible cause of post-bariatric surgery adrenal insufficiency could be malabsorption of factors that help in steroid biosynthesis such as bile, and therefore cholesterol levels, selenium, and other trace elements, and vitamin B5 and other vitamins. Additionally, weight loss causes re-setting of the hypothalamo-pituitary–adrenal axis [119,120].

The plasma cortisol profiles in patients with adrenal insufficiency can be almost similar before and after bariatric surgery. However, individual variations may exist [121]. However, in patients having persisting symptoms of adrenal insufficiency despite oral hydrocortisone treatment, the cortisol day curve may be abnormal due to malabsorption of oral hydrocortisone [119].

## 6. Summary

This review highlighted the importance of involving a Barocraniologist, or at least an Endocrinologist as part of the bariatric team in all stages of management of a patient who is a candidate for bariatric surgery. Both obesity and weight loss after bariatric surgery are associated with important metabolic and endocrine changes. Additionally, obesity is often the result of endocrine and metabolic imbalances. Hence, proper management of obesity is only possible when these endocrine and metabolic imbalances are appropriately managed. With changing lifestyles and growing cases of obesity and diabetes in countries like India, there is a felt need to incorporate a proper endocrine evaluation in obese patients.

## Figures and Tables

**Figure 1 medsci-08-00051-f001:**
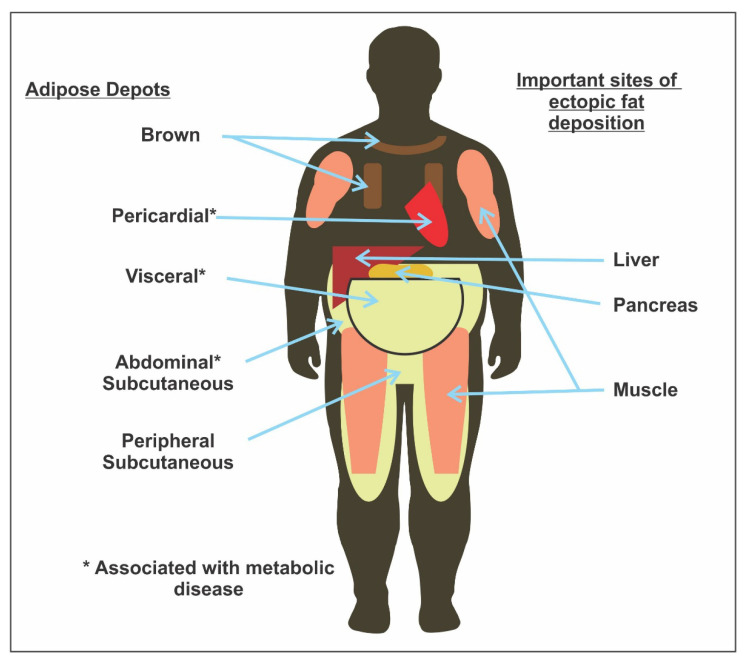
Sites of ectopic fat deposition [29].

**Figure 2 medsci-08-00051-f002:**
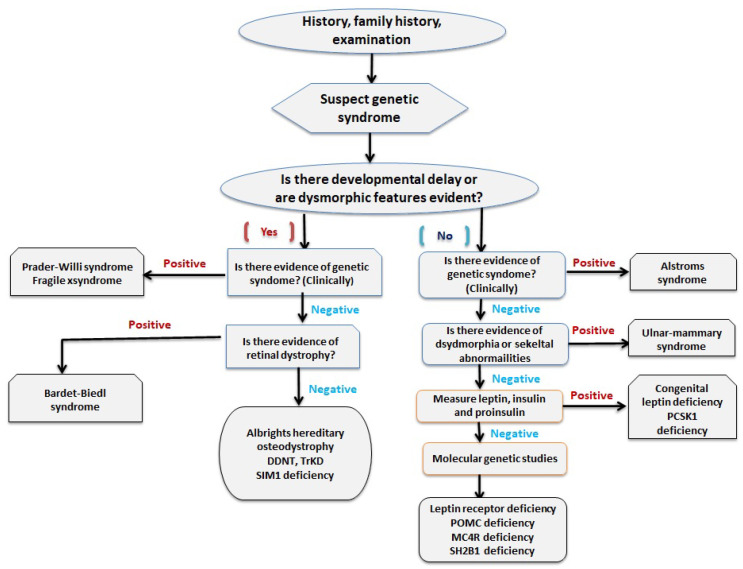
A flowchart showing the approach to diagnosing syndromic obesity [32]. Abbreviations: BDNF, brain derived neurotrophic factor; MC4R, melanocortin-4 receptor; POMC, proopiomelanocortin; SIM1, single-minded homolog 1; TrKB, tyrosine kinase B.

**Figure 3 medsci-08-00051-f003:**
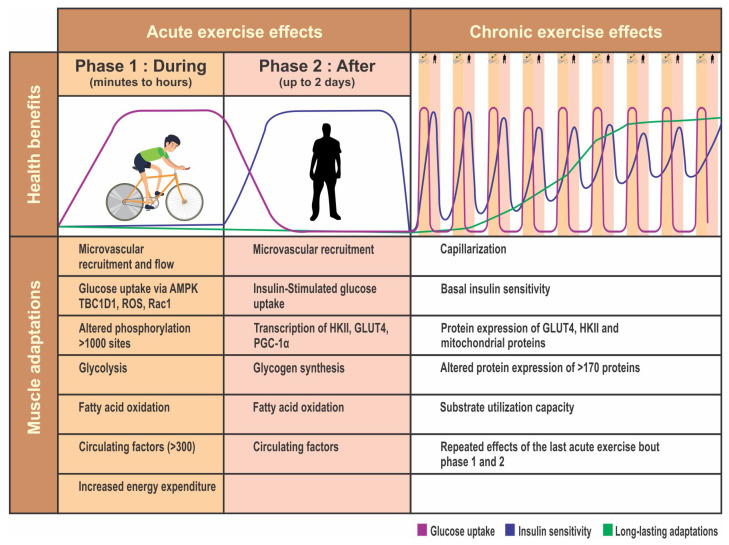
Benefits of exercise.

**Figure 4 medsci-08-00051-f004:**
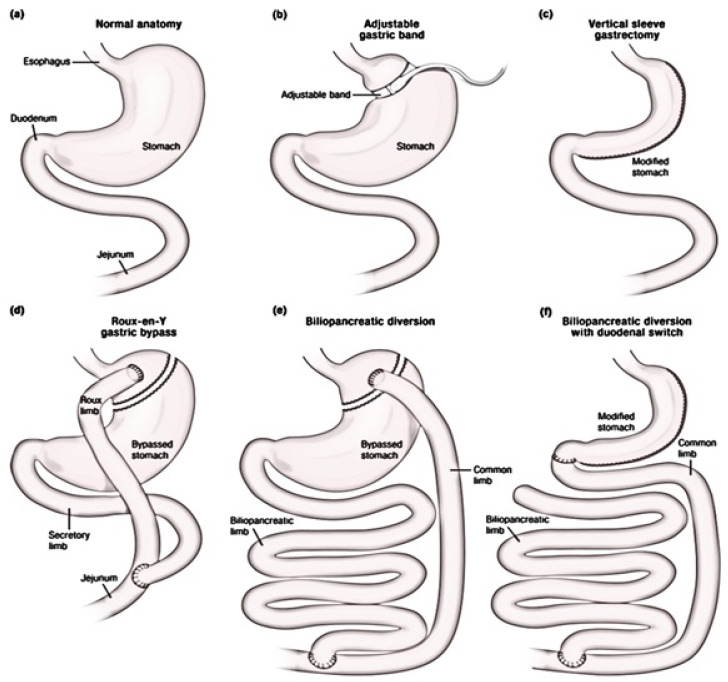
(**a**) Normal upper GI anatomy. Bariatric procedures: (**b**) Adjustable Gastric Banding (AGB); (**c**) Vertical Sleeve Gastrectomy (VSG); (**d**) Roux-en-Y gastric bypass (RYGB). (**e**) Biliary-Pancreatic Diversion. (**f**) Biliary-pancreatic Diversion with Duodenal Switch (BPD-DS) [73].

**Table 1 medsci-08-00051-t001:** Syndromic obesity [6,9].

Syndrome	Occurrence	Genetic Mutation	Features
Prader-Willi syndrome	1:10,000/1:15,000 births	*15q11 SRNPN* microdeletion	Newborn: Characteristic facies, small hands and feet, hypopigmentation hypotonia and failure to thrive Childhood: Short stature, hyperphagia, obesity, hypogonadism, delayed motor/cognitive development, sleep disturbances, and behavior abnormalities
Bardet–Biedel syndrome	1:125,000/1:175,000 births	*BBS* (1–12) chaperonin protein MKKS (Chr20)	Obesity in first year of life, mental retardation, dysmorphic extremities, retinal dystrophy or pigmentary retinopathy, hypogonadism. kidney anomalies
Albright’s hereditary osteodystrophy	1: 20,000/1:1,000,000 births	Autosomal dominant GNAS1 gene (20q13.2)	Short stature, obesity, round face, brachydactyly, subcutaneous calcification, dental and sensorineurat abnormalities Hormonal abnormalities: Generalizd hormonal resistance to PTH, TSH, GHRH and gonadotrophins; functional hypoparathyroidism seen on biochemical examination
Fragile X syndrome	1:2500 births	Unstable expansions of a *CGG* repeats—*FMR1* (fragile X mental retardation) gene- X chromosome	Macroorchidism, large ears, prominent jaw, hyperkinetic behavior, and mental retardation.
Wilson–Turner syndrome	Prevalence not known [10]	X-linked mutation	Truncal obesity, intellectual disability, gynecomastia, dysmorphic facial features, hypogonadism, and short stature
Cohen syndrome	Diagnosed in fewer than 1000 births worldwide	Chromosome 8q, and a novel gene, *COH1*	Dysmorphic extremities, microcephaly, prominent central incisors, retinal dystrophy, and cyclic neutropenia
Alstorm syndrome	Little more than 900 cases reported worldwide [11]	*2p14 ALSM1*	Progressive loss of vision and hearing, dilated cardiomyopathy, obesity, type 2 diabetes, and short stature

Abbreviations: GHRH, growth hormone-releasing hormone; PTH, parathyroid hormone; TSH, thyroid-stimulating hormone.

**Table 2 medsci-08-00051-t002:** Hormonal changes in obesity and after weight loss [12,13,15].

Hormone	Obesity	Proposed Pathophysiology in Obesity	Weight Loss/Fasting
Pituitary			
Prolactin			
Basal prolactin	N		N
Prolactin response to hypoglycemia	N/↓		N/↓
Prolactin response to TRH	N/↓		N
GH/IGF-I			
GH	N/↓	↓↑ GHRH, ↑ GH-BP, ↑ insulin, ↓ ghrelin, ↑ somatostatin	↑
GH production rate	↓		↑
GH metabolic rate	↑		
GH response to GHRH	↓		N/↓
GH response to hypoglycemia	↓		N/↓
IGF-I	N/↓	I ↑ GH sensitivity t ↑ intrahepatic triglyceride content	
Free IGF-I	↑		
IGFBP-3	N/↑		
HPA axis in obesity			
Basal cortisol (blood and salivary)	N	↑↑ CRH, ↑ adipose 11-HSD, ↓ CBG Altered suppression tests due to hyperactivity of the HPA axis	
Urinary free cortisol	N/↑		
Basal ACTH	N/↑	↑ CRH	
Cortisol production rate	↑		
Cortisol metabolic rate	↑		
17-OH-corticosteroids	↑		
HPA axis response to stimulatory tests in central obesity			
Cortisol after CRH	↑		N
ACTH after CRH	↑		↑
Cortisol after tetracosactide	↑		
Cortisol after hypoglycemia	↑		
Thyroid			
FT4	N/slight ↓	↑↑ disposal	N
T3	↑	Probably related to the amount of food intake and not to obesity itself.	↓
rT3	↓		↑
TSH	N/↑	↑↑ leptin and insulin ↑ peripheral T4 disposal	N
TSH after TRH	N/↑/↓		TRH dose decreases
Parathyroid			
PTH	N/↑	Secondary due to vitamin D deficiency	
	PTH ↑ calcium ↓ phosphate ↑	Pseudohypoparathyroidism Type 1a (Albright hereditary osteodystrophy)	
Pancreas/Gut			
Insulin	↑	Insulin resistance Insulinoma	↓ with decreased insulin resistance
Ghrelin	↓	Lack of ghrelin decrease after meals	↑
GLP-1	↓	↑ FFA, microbiota	↑
Gonads			
Testosterone (male)	↓	↓ SHBG, ↑ aromatase, ↓ GnRH	↑ testosterone and gonadotrophins, ↓ estradiol
Testosterone (female)	↑	Insulin resistance (PCOS) ↓ SHBG	↓ serum total testosterone, androsterone and sulfate dehydroepiandrosterone (DHEA)
LH/FSH(male)	↓	↑ oestrogens/androgens	↑ SHBG, LH, and FSH
LH/FSH(female)	↑ LH	Insulin resistance	↑ Serum estradiol, FSH and SHBG
Adrenal			
Renin	↑	↑ Sympathetic tone	
Aldosterone	↑	↑ Adipokines, renin- angiotensin, leptin	
Adipocyte			
Leptin	↑	↑ adipose mass, Leptin resistance	↓

Abbreviations: 11-HSD, 11β-hydroxysteroid dehydrogenase; ACTH, adrenocorticotropic hormone; CBG, corticosteroid-binding globulin; CRH, corticotropin-releasing hormone; FFA, free fatty acids; FSH, follicle-stimulating hormone; FT4, free thyroxine; GH-BP, growth hormone-binding protein; GHRH, growth hormone-releasing hormone; GLP, glucagon-like peptide; GnRH, gonadotropin-releasing hormone; HPA, hypothalamic–pituitary–adrenal axis; IGF, insulin-like growth factor; LH, luteinizing hormone; PCOS, polycystic ovary syndrome; PTH, parathyroid hormone; SHBG, sex hormone-binding globulin; TSH, thyroid-stimulating hormone. ↑ Increase, ↓ decrease.

**Table 3 medsci-08-00051-t003:** Endocrine evaluation in obesity [13,16,17,18].

Gland	Prevalence in Obesity	When to Assess	First Diagnostic Procedure	Other Mandatory Work Up in Obesity	Not Recommended in Obesity
Thyroid (Same TSH and thyroid hormone values should be used in patients with obesity as are used in normal population without obesity)	Severe hypothyroidism is rare but subclinical hypothyroidism is common	Thyroid function should be tested for all patients with obesity	TSH	free T4 and antibodies (anti-TPO) should be measured only if TSH is elevated	Routine FT3 in patients with elevated TSH Routine ultrasound of the thyroid gland (irrespective of thyroid function)
Adrenal (same normal values should be applied patients with obesity as are used in normal population without obesity)	Cushing’s disease or Cushing’s syndrome is rare	Central obesity Hypertension Type 2 diabetes Testing for hypercortisolism should be considered in patients going for bariatric surgery	1 mg ODST	24 h urine cortisol or late-night salivary cortisol in patients with positive 1 mg overnight dexamethasone suppression test Imaging (find the cause/source) and ACTH in patients with confirmed hypercortisolism	Routine testing for hypercortisolism
	Drug-induced adrenal dysfunction (e.g., lithium, anti-depressants, antipsychotics, glucocorticoids, etc.) is common	Biochemical testing should be performed in patients with clinical suspicion of hypercortisolism; those undergoing bariatric surgery, or having psychiatric disorders	8:00 a.m. cortisol		Testing for hypercortisolism in patients using corticosteroids
Male Gonad (Use age-specific reference ranges for testosterone	Androgen deficiency is common	Severe obesity Symptoms and signs of hypogonadism In elderly male with impaired social and mental health, less energy	LH, FSH, fasting morning testosterone	Total and free testosterone (or calculated), SHBG in patients with clinical features of hypogonadism	Routine biochemical testing for hypogonadism unless key clinical symptoms/signs of hypogonadism and in elderly with impaired social/mental health, less energy
Female Gonad	Androgen excess is common	Central obesity Irregular menses Hirsutism Acanthosis nigricans chronic anovulation/infertility	LH, FSH, estradiol, testosterone	Total testosterone, SHBG, Δ 4 androstenedione, 17-hydroxyprogesterone and prolactin in patients with menstrual irregularities (assess in early follicular phase if menstrual cycle is predictable)	Routine testing for gonadal dysfunction
		Clinical features of PCOS	Total testosterone, free T, Δ 4 androstenedion, SHBG and blood glucose	Ovarian morphology	
	Premature ovarian failure is uncommon	Secondary amenorrhea Vasomotor symptoms	LH, FSH, estradiol	Progesterone and prolactin in patients with anovulation	
	Physiological ovarian failure in menopause is common	Vaginal mucosal atrophy	LH, FSH, estradiol		
Pituitary	GH deficiency is rare	Hypothalamic or pituitary disease, pituitary or hypothalamic surgery or radiation therapy		IGF1/GH using a dynamic test only in patients with suspected hypopituitarism	Routine testing for IGF1/GH
	Hypopituitarism is rare	Suspicion of hypothalamic obesity Surgery or radiotherapy in pituitary region	FT4 TSH LH FSH (testosterone or estradiol) GH IGF-1 PRL ACTH stimulation test GH stimulation test		
	Acquired hypothalamic obesity (hypothalamic lesions or, tumors) is rare	Severe hyperphagia Possible multiple endocrine abnormalities	Brain CT/MRI		
Parathyroid	Pseudohypoparathyroidism Type 1a (Albright hereditary osteodystrophy) is rare	Short stature, short fourth metacarpal bones, obesity, sc calcifications, developmental delay	PTH ↑ calcium ↓ phosphate ↑		Routine testing for hyperparathyroidism or Vitamin D deficiency
Hypothalamus obesity	Hypothalamic obesity associated with Genetic Syndromes is very rare	Hypogonadism (hypogonadism or hypergonadotropic) or variable gonadal function. dysmorphic syndrome, mental and grow retardation	Leptin (leptin resistance); genetic testing	Routine testing of hormones such as leptin and ghrelin in patients with suspicion of syndromic obesity	

Abbreviations: ACTH, adrenocorticotropic hormone; FSH, follicle-stimulating hormone; FT4, free thyroxine; GH, growth hormone; IGF, insulin-like growth factor; LH, luteinizing hormone; MC4R, melanocortin receptor 4; ODST, overnight dexamethasone suppression test; PCSK, proprotein convertase subtilisin/kexin; PTH, parathyroid hormone; sc, subcutaneous; TSH, thyroid-stimulating hormone. ↑ Increase, ↓ decrease.

**Table 4 medsci-08-00051-t004:** Body mass index classification of obesity.

	Asian Body Mass Index Classification in Adults [19,20]	Body Mass Index Classification for Adults by World Health Organization [20]
Classification	Body mass index (kg/m^2^)	Body mass index (kg/m^2^)
Underweight	<18.5	<18.5
Normal weight	18.5–22.9	18.5–24.9
Overweight	23–24.9	25.0–29.9
Obesity class I	25–29.9 kg/m^2^	30.0–34.9
Obesity class II	30–34.9 kg/m^2^	35.0–39.9
Obesity class III	≥35 kg/m^2^	≥40

**Table 5 medsci-08-00051-t005:** Comparing the predictive value of waist circumference, waist-to-hip ratio, and body mass index for individual adipose tissue compartments [28].

Non-Nested Models	WC vs. WHR	WC vs. BMI	BMI vs. WHR
	*p* Values		
Retroperitoneal ATM	0.670	0.045	0.115
Intraperitoneal ATM	0.285	0.042	0.544
Subcutaneous anterior ATM	<0.001	0.280	0.036
Subcutaneous posterior ATM	<0.001	0.759	<0.001

Abbreviations: WC, waist circumference; WHR, waist-to-hip ratio; BMI, body mass index; ATM, adipose tissue mass.

**Table 6 medsci-08-00051-t006:** Pharmacotherapy of obesity [51].

Drug	Mechanism of Action	Common Adverse Effects	Warnings	Contraindications
Phentermine	Sympathomimetic amine	Headache, dizziness, fatigue, dry mouth, constipation, upper respiratory tract-like symptoms and hypoglycaemia in patients with diabetes mellitus	Rare cases of primary pulmonary hypertension or serious regurgitant cardiac valvular disease	Cardiovascular disease, uncontrolled hypertension, agitated states, history of drug use, hyperthyroidism, glaucoma or MAGI use within 14 days
Orlistat	Pancreatic and gastric lipase inhibitor	Flatulence, bloating and diarrhoea	Malabsorption of fat-soluble vitamins A, D, E and K	Malabsorption syndrome or cholestasis
Phentermine/topiramate ER	Combination of sympathomimetic amine, anorectic and ER antiepileptic drug	Peripheral neuropathy (usually transient), dyspepsia, insomnia, constipation and dry mouth	Teratogenicity (risk of cleft palate and/or cleft lip), suicidal ideation, changes in memory or concentration, metabolic acidosis, hypokalaemia	Glaucoma, hyperthyroidism or MAOI use within 14 days
Lorcaserin (currently is banned/debarred due to increased cancer risk)	5-HT_2c_ receptor agonist	Headache, dizziness, fatigue, dry mouth, constipation, upper respiratory tract-like symptoms and hypoglycaemia in patients with diabetes mellitus	Serotonin syndrome or neuroleptic malignant syndrome; safety unknown with coadministration of serotonin or antidopaminergic agents, cognitive impairment, suicidal ideation and valvular heart disease (though not statistically significant in clinical trials)	Pregnancy
Naltrexone SR	Combination opioid antagonist and aminoketone antidepressant	Nausea, constipation, headache, vomiting, dizziness, dry mouth and diarrhoea	Suicidal ideation, decrease in seizure threshold, acute angle glaucoma, hepatotoxicity to naltrexone component and increase in HR and/or BP	Uncontrolled hypertension, seizure disorders, anorexia nervosa or bulimia, chronic opioid use, MAOI use within 14 days, abrupt discontinuation of alcohol or seizure medications
Liraglutide (3 mg)	GLP1 analogue	Hypoglycemia, nausea, diarrhea, vomiting, constipation, dyspepsia, decreased appetite, headache, fatigue, dizziness, abdominal pain, and increased lipase	Serious hypoglycemia, thyroid C-cell tumors, acute gallbladder disease, acute pancreatitis, hypersensitivity reactions.	Personal or family history of MTC or MEN syndrome type 2, hypersensitivity to liraglutide, Pregnancy

Abbreviations: 5-HT_2c_, 5-hydroxytryptamine receptor 2C; BP, blood pressure; ER, extended release; GLP1, glucagon-like peptide 1; HR, heart rate; MAOI, monoamine oxidase inhibitor; MEN, multiple. Endocrine neoplasia; MTC, medullary thyroid carcinoma; OTC, over the counter; SR, sustained release.

**Table 7 medsci-08-00051-t007:** Medications prescribed and their treatment profile [56,57].

Medications	Weight Profile
Antidiabetes	GLP-1 analogs (eg, exenatide, albiglutide, dulaglutide, semaglutide and liraglutide) or SGLT-2 inhibitors (dapagliflozin, empagliflozin and canagliflozin) promote weight loss Basal insulin causes less weight gain than other insulin types
Antihypertensives	Angiotensin-converting enzyme (ACE) inhibitors, angiotensin receptor blockers (ARBs), and calcium channel blockers should be preferred over β-adrenergic blockers which cause weight gain
Antidepressants	Paroxetine, amitryiptyline, mirtazapine, and nortriptyline are linked to weight gain Bupropion causes weight loss
Antipsychotics	Clozapine and olanzapine have a greater likelihood for weight gain, while ziprasidone appears to have the lowest risk for weight gain
Antiepileptics	Felbamate, topiramate and zonisamide may be preferred over other antiepileptics as they cause weight loss
Antihistamines	Choose one with less sedation
Antiretroviral	Most antiretrovirals cause weight gain, weight monitoring is important

**Table 8 medsci-08-00051-t008:** Gut hormones and their actions and effects; three common bariatric procedures on gut hormone regulation [77].

Hormone	Organ/Cell	Mechanism of Action	Levels after Bariatric Procedure		
			GB	SG	RYGB
CCK	Duodenum/I cells	Stimulates digestion of fats and proteins	No data	Increase	Increase
FGF19	Ileum	Regulation of glucose and lipid metabolism. Increases energy expenditure	Increase	Increase	Increase
FGF21	Liver	Fatty acid oxidation, improves insulin sensitivity and increases energy expenditure	No data	Decrease	No change
Gastrin	Stomach/G cells	Increases HCl production	No change	No change/increase	Decrease
		Promotes satiety			
Ghrelin	Stomach/G cells	Increases appetite	No change/increase	Conflicting data	Conflicting data
		Enhances gastric emptying GI motility and GH secretion			
GLP-1	Ileum/L cells	Causes the incretin effect	No change/increase	Increase	Increase
		Increases insulin sensitivity and production. Delays gastric emptying. Enhances satiety.			
GLP-2	Ileum/L cells	Causes gut hypertrophy. Alters GI motility	No change	Increase	Increase
Glucagon	Pancreas/A cells	Promotes glucogenolysis and gluconeogenesis	No change	Decrease	Conflicting data
Gustducin	Stomach/specialized lining cells	Enhances GLP-1 secretion	No data	No data	Increase
Insulin	Pancreas/B cells	Regulates metabolism of carbs, fat and protein. Promotes absorption glucose from the blood	No change/increase	Increase	Increase
Obestatin	Stomach/epithelial cells	Promotes satiety	No data	Decrease	Increase
Secretin	Duodenum/S cells	Reduces gastric and duodenal motility. Enhances insulin release	No data	No data	No change/decrease
VIP	Enteric and parasympathetic nerves	Promotes hormone secretion by the brain, gut and pancreas Increases the secretion of water and electrolytes. Reduces HCl secretion	No data	No data	No data

Abbreviations: CCK, cholecystokinin; Conflicting data, no change, increase, decrease; FGF19, fibroblast growth factor 19; FGF21, fibroblast growth factor 21; GB, Gastric banding; GIP, glucose-dependent insulinotropic polypeptide; GLP-1, glucagon-like peptide 1; GLP-2, glucagon-like peptide 2; PP, pancreatic polypeptide; PYY, polypeptide YY; RYGB: Roux-en-Y gastric bypass; SG, Sleeve gastrectomy; VIP, vasoactive intestinal polypeptide.

**Table 9 medsci-08-00051-t009:** Normative range of TSH cut-offs proposed on the basis of body mass index [118].

BMI Category	TSH-Cut-Off
<20 kg/m^2^	0.6 to 4.8 µUI/mL
20–24.9 kg/m^2^	0.6 to 5.5 µUI/mL
25–29.9 kg/m^2^	0.6 to 5.5 µUI/mL
3–39.9 kg/m^2^	0.5 to 5.9 µUI/mL
>40 kg/m^2^	0.7 to 7.5 µUI/mL

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
