# Peer review of "Barocrinology: The Endocrinology of Obesity from Bench to Bedside"

_medsci, 2020, doi:10.3390/medsci8040051_

Round 1

Reviewer 1 Report

I read with interest the manuscript of Kalra et al. regarding the main endocrine and metabolic elements of weight physiology and pathology.

The manuscript is interesting and a very comprehensive narrative review. I have only some minor points:

- in table 3, the use of “Syndromic obesity” in the "gland" cell is not appropriate and it should be replaced with another term 

- in table 6, why didn't the authors also include liraglutide 3 mg?

- in table 7, I am not sure that metformin can effectively provide weight loss, usually it is neutral on weight 

Author Response

Reviewer 1:

  1. I read with interest the manuscript of Kalra et al. regarding the main endocrine and metabolic elements of weight physiology and pathology.

The manuscript is interesting and a very comprehensive narrative review. I have only some minor points:

We thank the reviewer for the positive comments.

  1. - in table 3, the use of “Syndromic obesity” in the "gland" cell is not appropriate and it should be replaced with another term

We thank the reviewer for pointing this error. We have now replaced the term “Syndromic obesity” to “Hypothalamus” in Table 3.

  1. - in table 6, why didn't the authors also include liraglutide 3 mg?

We agree with this point and have now included the same as below

Drug

Mechanism of action

Common adverse effects

Warnings

Contraindications

Liraglutide (3 mg)

GLP-I Analogue

Hypoglycemia, nausea, diarrhea, vomiting, constipation, dyspepsia, decreased appetite, headache,

fatigue, dizziness, abdominal pain, and

Serious hypoglycemia, thyroid C-cell tumors, acute gallbladder disease, acute pancreatitis, hypersensitivity reactions.

Personal or family history of MTC or MEN syndrome type 2, hypersensitivity to liraglutide, Pregnancy y

Added to abbreviations: MEN, multiple endocrine neoplasia; MTC, medullary thyroid carcinoma;

  1. - in table 7, I am not sure that metformin can effectively provide weight loss, usually it is neutral on weight 

We agree and thank the reviewer for this comment.

We have removed this from the table.

Medications

Weight profile

Antidiabetes

Metformin, GLP-1 analogs (eg, exenatide, albiglutide, dulaglutide, semaglutide and liraglutide) or SGLT-2 inhibitors (dapagliflozin, empagliflozin and canagliflozin) promote weight loss

Basal insulin causes less weight gain than other insulin types

Reviewer 2 Report

The review presented by Sanjay Kalra, et al. entitled: “Barocrinology: The Endocrinology of Obesity from Bench to Bedside” and submitted to revision, tackles very interesting aspects related to obesity bariatric and pharmacological management and the endocrinology. I think this article should be accepted for publication with minimal changes since it summarizes and contributes significantly to overall knowledge in this disease.  The paper is well-written although, changes minimal must be included to sustain the quality of the manuscript.

Table 2: check the table, there is a mistake

Line 87: check the paragraph.

Table 3: it is based on an only reference and should be presented a little more clearly because it leads to error

Line 194: Say it is MNT

Line 252 and 253: I think is better to say antidiabetics and antihistaminics

Author Response

Reviewer 2:

  1. The review presented by Sanjay Kalra, et al. entitled: “Barocrinology: The Endocrinology of Obesity from Bench to Bedside” and submitted to revision, tackles very interesting aspects related to obesity bariatric and pharmacological management and the endocrinology. I think this article should be accepted for publication with minimal changes since it summarizes and contributes significantly to overall knowledge in this disease.  The paper is well-written although, changes minimal must be included to sustain the quality of the manuscript.

       We thank the reviewer for the positive comments.

  1. Table 2: check the table, there is a mistake

       We have modified the errors in the table as below:

Table 2. Hormonal changes in obesity and after weight-loss9,10,113.

Hormone

Obesity

Proposed pathophysiology in obesity

Weight-loss /Fasting

Pituitary

Prolactin

Basal prolactin

N

N

Prolactin response to hypoglycemia

 N/

N/

Prolactin response to TRH

N/

N

GH/IGF-I

GH

N/

↓↑GHRH, ↑GH-BP, ↑insulin, ↓ghrelin, ↑somatostatin

GH production rate

GH metabolic rate

GH response to GHRH

N/

GH response to hypoglycemia

N/

IGF-I

N/

I↑GH sensitivity

t↑intrahepatic triglyceride content

Free IGF-I

IGFBP-3

N/

HPA axis in obesity

Basal cortisol (blood and salivary)

N

↑↑CRH, ↑ adipose 11-HSD, ↓ CBG

Altered suppression tests due to hyperactivity of the HPA axis

Urinary free cortisol

N/

Basal ACTH

N/

↑CRH

Cortisol production rate

Cortisol metabolic rate

17-OH-corticosteroids

HPA axis response to stimulatory tests in central obesity

Cortisol after CRH

N

ACTH after CRH

Cortisol after tetracosactide

Cortisol after hypoglycemia

Thyroid

FT4

N/slight

↑↑disposal

N

T3

Probably related to the amount of food intake and not to obesity itself. (kkoris ref nos 8-12)

rT3

TSH

N/

↑↑leptin and insulin

↑ peripheral T4 disposal

N

TSH after TRH

N/↑/↓

TRH dose decreases

Parathyroid

PTH

N/

Secondary due to vitamin D deficiency

PTH ↑

calcium ↓ phosphate ↑

Pseudohypoparathyroidism Type 1a

(Albright hereditary

osteodystrophy)

Pancreas/Gut

Insulin

IInsulin resistance

  Insulinoma nsulinomae istanc

↓ with decreased insulin resistance

Ghrelin

Lack of ghrelin decrease after meals

GLP-1

↑ FFA, microbiota

Gonads

Testosterone (male)

↓SHBG, ↑ aromatase, ↓GnRH

↑ testosterone and gonadotrophins, ↓ estradiol

Testosterone (female)

Insulin resistance (PCOS) ↓ SHBG

↓ serum total testosterone, androsterone and sulfate dehydroepiandrosterone (DHEA)

LH/FSH(male)

↑oestrogens/androgens

↑ SHBG, LH, and FSH

LH/FSH(female)

LH

Insulin resistance

Serum estradiol, FSH and SHBG

Adrenal

Renin

↑ Sympathetic tone

Aldosterone

↑ Adipokines, renin- angiotensin, leptin

Adipocyte

Leptin

 ↑ adipose mass, Leptin resistance

Abbreviations: 11-HSD, 11β-hydroxysteroid dehydrogenase; ACTH, adrenocorticotropic hormone; CBG, corticosteroid-binding globulin; CRH, corticotropin-releasing hormone; FFA, free fatty acids; FSH, follicle-stimulating hormone; FT4, free thyroxine; GH-BP, growth hormone-binding protein; GHRH, growth hormone-releasing hormone; GLP, glucagon-like peptide; GnRH, gonadotropin-releasing hormone; HPA, hypothalamic–pituitary–adrenal axis; IGF, insulin-like growth factor; LH, luteinizing hormone; PCOS, polycystic ovary syndrome; PTH, parathyroid hormone; SHBG, sex hormone-binding globulin; TSH, thyroid-stimulating hormone

  1. Line 87: check the paragraph.

       We have now corrected the same.

       3.1. Endocrine condition: laboratory evaluation: dos and don’ts

       Laboratory evaluation for endocrinopathies in a patient with obesity

     4. Table 3: it is based on an only reference and should be presented a little   more clearly because it leads to error

       Thanks for the feedback. All the references used in the table have  been mentioned in the table header

      Table 3. Endocrine evaluation in obesity10,12–14.

Gland

Prevalence in obesity

When to assess

First diagnostic procedure

Other possible work up in obesity

Not recommended in obesity

Thyroid (Same TSH and thyroid hormone values should be used in patients with obesity as are used in normal population without obesity)

Severe hypothyroidism is rare but subclinical hypothyroidism is common. It is important to render the patient Euthyroid prior to embarking on weight loss therapy to enhance maximum effect of treatment

Thyroid function should be tested for all patients with obesity

TSH

free T4 and antibodies (anti-TPO) should be measured only if TSH is elevated

Routine FT3 in patients with elevated TSH

Routine ultrasound of the thyroid gland (irrespective of thyroid function)

Adrenal (same normal values should be applied patients with obesity as are used in normal population without obesity)

Cushing’s disease or Cushing’s syndrome is rare

But exogenous cushings syndrome is common.

Central obesity

Hypertension

Type 2 diabetes

Purplish striae

Facial plethora

Easy bruisability

Testing for hypercortisolism should be considered in patients going for bariatric surgery

8 am cortisol (to rule out exogenous steroid intake)       1 mg ODST

24-h urine cortisol or late-night salivary cortisol in patients with positive 1 mg overnight dexamethasone suppression test

Imaging (find the cause/source) and ACTH in patients with confirmed hypercortisolism

Routine testing for hypercortisolism

Drug-induced adrenal dysfunction

(e.g. anti-depressants,

antipsychotics, glucocorticoids, etc) is common

Biochemical testing should be performed in patients with clinical suspicion of hypercortisolism; those undergoing bariatric surgery, or having psychiatric disorders

8 am cortisol

Testing for exogenous hypercortisolism in patients using corticosteroids

Male Gonad

(Use age-specific reference ranges for testosterone

Androgen deficiency is common

Severe obesity

Symptoms and

Signs of hypogonadism

In elderly male with impaired social and mental health, less energy

LH, FSH, fasting morning testosterone

Total and free testosterone (or calculated), SHBG in patients with clinical features of hypogonadism

Routine biochemical testing for hypogonadism

unless key clinical symptoms/signs of hypogonadism and in elderly with impaired social/mental health, less energy

Female Gonad

Androgen excess is common

Central obesity

Irregular menses

Hirsutism

Acanthosis nigricans chronic anovulation/infertility

LH, FSH, estradiol, testosterone

Total testosterone, SHBG, Δ 4androstenedione, 17-hydroxyprogesterone and prolactin in patients with menstrual irregularities (assess in  early follicular phase if menstrual cycle is predictable)

Routine testing for gonadal dysfunction

Clinical features of PCOS

Total testosterone, free T, Δ 4androstenedion, SHBG and blood glucose

Ovarian morphology

Premature ovarian failure is uncommon

Secondary amenorrhea Vasomotor

symptoms

LH, FSH, estradiol

Progesterone and prolactin in patients with anovulation

Physiological ovarian failure in menopause is common

Vaginal mucosal atrophy

LH, FSH, estradiol

Pituitary

GH deficiency is rare

Hypothalamic or pituitary disease,

pituitary or hypothalamic

surgery or radiation therapy

IGF1/GH using a dynamic test only in patients with suspected hypopituitarism

Routine testing for IGF1/GH

Hypopituitarism is rare

Suspicion of hypothalamic obesity

Surgery or radiotherapy in

pituitary region

FT4 TSH LH FSH (testosterone or

estradiol)

GH IGF-1 PRL

ACTH stimulation test

GH stimulation test

Acquired hypothalamic obesity

(hypothalamic lesions or, tumors) is rare

Severe hyperphagia

Possible multiple endocrine

abnormalities

Brain CT/MRI

Parathyroid

Pseudohypoparathyroidism Type 1a

(Albright hereditary

osteodystrophy) is rare

Short stature, short fourth

metacarpal bones, obesity, sc

calcifications, developmental

delay

PTH ↑

calcium ↓ phosphate ↑

Routine testing for hyperparathyroidism or Vitamin D deficiency

Hypothalamus

Hypothalamic obesity associated with

Genetic Syndromes is very rare

Hypogonadism (hypogonadism or

hypergonadotropic) or variable

gonadal function. dysmorphic

syndrome, mental and grow

retardation

Leptin (leptin resistance);genetic testing

Routine testing of hormones such as leptin and ghrelin in patients with suspicion of syndromic obesity

Abbreviations: ACTH, adrenocorticotropic hormone; FSH, follicle-stimulating hormone; FT4, free thyroxine; GH, growth hormone; IGF, insulin-like growth factor; LH, luteinizing hormone; MC4R, melanocortin receptor 4; ODST, overnight dexamethasone suppression test; PCSK, proprotein convertase subtilisin/kexin; PTH, parathyroid hormone; sc, subcutaneous; TSH, thyroid-stimulating hormone

    5.Line 194: Say it is MNT

   We have modified this line as suggested

   5.1. Potential endocrine/metabolic impact of formula MNT

   5.1 Medical Nutrition Therapy (MNT)

  1. Line 252 and 253: I think is better to say antidiabetics and antihistaminics

     We have modified the same

Common medications likely to cause weight gain include antipsychotics, antidiabetics, antihypertensives, antidepressants, antihistaminics etc. Medications, that are either weight neutral or have weight loss as a side effect, should be preferred over medications that have weight gain as a side effect

These have been added. Dr Punit-these are not to be included in the journal comment reply

In addition if could please do two small changes –

Replace reference 19 with this  - Kapoor N, Jiwanmall SA, Nandyal MB, Kattula D, Paravathareddy S, Paul TV, Furler J, Oldenburg B, Thomas N. Metabolic Score for Visceral Fat (METS-VF) Estimation - A Novel Cost-Effective Obesity Indicator for Visceral Adipose Tissue Estimation. Diabetes Metab Syndr Obes. 2020 Sep 16;13:3261-3267. doi: 10.2147/DMSO.S266277. PMID: 32982356; PMCID: PMC7507406.

For line 101-103, if could add the following two references –

Atri A, Jiwanmall SA, Nandyal MB, Kattula D, Paravathareddy S, Paul TV, Thomas N, Kapoor N. The Prevalence and Predictors of Non-alcoholic Fatty Liver Disease in Morbidly Obese Women - A Cross-sectional Study from Southern India. Eur Endocrinol. 2020 Oct;16(2):152-155. doi: 10.17925/EE.2020.16.2.152. Epub 2020 Oct 6. PMID: 33117448; PMCID: PMC7572172.

Ramasamy S, Joseph M, Jiwanmall SA, Kattula D, Nandyal MB, Abraham V, Samarasam I, Paravathareddy S, Paul TV, Rajaratnam S, Thomas N, Kapoor N. Obesity Indicators and Health-related Quality of Life - Insights from a Cohort of Morbidly Obese, Middle-aged South Indian Women. Eur Endocrinol. 2020 Oct;16(2):148-151. doi: 10.17925/EE.2020.16.2.148. Epub 2020 Oct 6. PMID: 33117447; PMCID: PMC7572161.
